# Mechanism of Functional Compound Fruit Drinks in Regulating Serum Metabolism in Constipated Mice

**DOI:** 10.3390/ijms26020702

**Published:** 2025-01-15

**Authors:** Quanhong Lu, Yali Shi, Xin Wen, Lulu Zhu, Longteng Zhang, Kexue Zhu, Jun Cao, Chuan Li

**Affiliations:** 1Key Laboratory of Food Nutrition and Functional Food of Hainan Province, School of Food Science and Engineering, Hainan University, Haikou 570228, China; 15808019423@163.com (Q.L.); okthen1@163.com (Y.S.); 23210832000005@hainanu.edu.cn (X.W.); zhanglongteng@hainanu.edu.cn (L.Z.); juncaoyd2007@126.com (J.C.); 2Spice and Beverage Research Institute, Chinese Academy of Tropical Agricultural Sciences, Wanning 571533, China; zhukexue163@163.com

**Keywords:** compound fruit drink, constipation, serum metabolism

## Abstract

A compound fruit drink (CFD) is a functional beverage containing fruits, Chinese herbal medicine, and prebiotic fructose. Previous studies have shown the effect of a CFD on alleviating constipation and its impact on gut microbiota. However, a comprehensive analysis has not been reported in regard to the serum metabolism of CFDs. This study established a mouse constipation model, using loperamide hydrochloride. Herein, based on UHPLC–QTOF/MS analysis, 93 differential metabolites (mainly including phosphoglycerides and amino acid derivatives) among the groups of mice were identified. After CFD treatment, the content of phosphatidylethanolamine, amino acid derivatives (including N-Acetyl-L-aspartate, L-Norleucine, and cis-4-Hydroxy-D-proline), and fumarate increased, while that of esters decreased. Pathway enrichment analysis revealed that the CFD mitigated constipation by modulating nine metabolic pathways, which encompass glycerophospholipid metabolism, the tricarboxylic acid (TCA) cycle, pyruvate metabolism, and tyrosine metabolism. Notably, the glycerophospholipid metabolic pathway was identified as the most pertinent. Collectively, the results provide new ideas for developing functional foods that nourish the intestines and relieve constipation.

## 1. Introduction

Constipation is a prevalent issue in modern life, mainly manifested as persistent difficulty and reduced frequency of defecation (less than three times a week), dry stools, long bowel movements, and incomplete bowel movements [1,2]. Today, an estimated 12–14% of the global population experiences constipation [3]. In addition, constipation affects the patients’ physiological functions and reduces their quality of life, leading to increased psychological stress [1]. Frequent or long-term constipation not only causes metabolic disorders and allergic bowel syndrome, but also other related diseases [4].

Currently, the primary treatments for constipation include dietary fiber or laxatives, biofeedback therapy, and surgery [5]. However, the long-term use of medication can lead to adverse side effects and dependence in patients [6]. Although biofeedback therapy has good therapeutic effects, the safety and long-term impact of this constipation treatment still needs to be confirmed using large samples and long-term studies. Severe constipation is treated by surgery, but there are certain complications and recurrence rates after surgery, and other treatment methods are still needed [7]. A report showed that a kombucha-based drink enriched with inulin and vitamins improved constipation in women with irritable bowel syndrome, increased the frequency of their daily bowel movements, and improved stool consistency [8]. Therefore, developing safe and effective functional foods has practical significance for alleviating constipation.

Studies have shown that plant-based foods, especially fruits, contain phytochemicals that are a food source for gut microbiota. Microorganisms living in the lower digestive tract help break down plant chemicals and the bioactive substances in fruits are neuroprotective, antioxidant, and anti-inflammatory agents. Fruits also enhance intestinal peristalsis, facilitate the growth of beneficial bacteria, such as lactic acid bacteria and bifidobacteria, and help decrease the production of harmful bacteria and toxins [9]. Furthermore, they contribute to increasing the levels of short-chain fatty acids in the intestines and the density of tissue cells, which aids in lubricating the intestines and facilitating bowel movements [9]. Gearry et al. [10] have demonstrated that the consumption of kiwifruit alleviated constipation. It substantially increased the frequency of bowel movements in individuals suffering from constipation and promoted gastrointestinal comfort. Chinese medicinal materials with the same origin as food improved the ecological environment in the intestine, promoted intestinal peristalsis, and effectively prevented and improved constipation. Deng et al. showed that traditional Chinese herbal beverages increased the number of neurotransmitters, promoted intestinal peristalsis, and inhibited the expression of vasoactive intestinal peptides and intestinal peristalsis in rat serum, thereby relieving constipation [11].

Functional foods hold significant potential for alleviating constipation. Functional beverages contain multiple nutrients (amino acids, vitamins, antioxidants, etc.) and meet the needs of certain groups of people [12]. Researchers have found that consuming multiple foods together that regulate intestinal motility had a synergistic effect, which was generally better than consuming them separately [13]. Lu et al. reported that the combination of probiotics, konjac glucomannan, and Prunus persica extract effectively alleviated constipation [14]. At present, research has been conducted on the effect of CFDs on improving constipation and its impact on gut microbiota, but the regulatory mechanism of serum metabolites is not yet clear, which is crucial for elucidating the mechanism of functional composite drinks in improving constipation in mice [15]. 

This study prepared a CFD by mixing fruit powder (blueberries, snow lotus fruits, etc.) with traditional Chinese medicine liquids (licorice, tangerine peel, etc.) that were of the same origin as the food and medicine. A mouse constipation model was established using loperamide and the serum metabolites were identified using ultra-high performance liquid chromatography quadrupole time-of-flight mass spectrometry (UPLC–QTOF/MS). The metabolic pathways associated with constipation were identified through pathway enrichment analysis and the small-molecule metabolic mechanism of the CFD in alleviating constipation in mice was explored. This study aims to provide some references for exploring the use of composite functional beverages to alleviate constipation, which is of great significance for the development of food and drug homologous raw materials.

## 2. Results and Discussion

### 2.1. The Effect of the CFD on Serum Metabolites

Principal Component Analysis (PCA) reduces the dimensionality of the data, preserves the most important directions of change in the data, and is commonly used in the comprehensive analysis of multidimensional data clustering [16]. As shown in Figure 1A,D, the PCA score plots of the serum metabolites in positive and negative modes for the control, model, and CFD treatment groups are presented. In the scatter plot, PC1 and PC2 accounted for 27.1 and 20.7% (positive mode), 20.4 and 15% (negative mode), respectively. The serum from each group showed good separation effects in the positive and negative modes. This indicated that there were significant changes in the metabolites in mouse serum after CFD treatment compared to the control group and the relative separation between the groups indicated significant differences in the small-molecule metabolites. The OPLS-DA model demonstrates the differences in the small-molecule metabolites between the groups better than the PCA model and is further used to investigate the effect of the CFD on small-molecule metabolism in constipated mice. The model reflected the relationship between the metabolites and sample categories, removed the irrelevant variables in the classification information, and improved the accuracy of sample classification [17]. The results demonstrated that the OPLS-DA model exhibited superior clustering performance compared to the PCA model. The R^2^Y and Q^2^ in the OPLS-DA model in positive mode were 0.904 and 0.622, and those in negative mode were 0.949 and 0.761, respectively (Figure 1B,E).

The significance of the OPLS-DA model was evaluated through 200 repeated replacement experiments. The Q^2^ regression line intercept of the permutation test was negative and the original intercept value was higher than the intercept values of all the permutation vectors (Figure 1C,F). This indicated that the OPLS-DA model had good predictive ability and effectiveness [18].

### 2.2. The Effect of the CFD on Intergroup Metabolites

Differential metabolites were identified through the analysis of the degree of variation between the two groups (MC vs. NC, CFD-L vs. MC, CFD-M vs. MC, CFD-H vs. MC). The S-plot and S-line represent the differences in the substance content between the groups. The results showed that the MC group and NC group in the OPLS-DA model were clustered clearly into two groups (Figure 2A). Similarly, the data for the CFD-L, CFD-M, CFD-H, and MC groups in the OPLS-DA model were separated unambiguously (Figure 2C,E,G). This showed that the metabolites in the mice with constipation induced by loperamide changed and that the CFD treatment could intervene in this phenomenon. The R^2^Y and Q^2^ of the MC vs. NC group, the CFD-L vs. MC group, the CFD-M vs. MC group, and the CFD-H vs. MC group, were 0.999 and 0.94, 0.999 and 0.866, 0.999 and 0.946, 0.999 and 0.953, respectively, which pointed out the high goodness-of-fit prediction ability of the model [19]. R^2^Y represents the model’s fitness and Q^2^ represents the model’s predictive ability. The Q^2^ was > 0.5, demonstrating that the PCA model was reliable. The left side of the S-plot represents substances with a downward trend, while the right side represents substances with an upward trend, and the difference in the metabolites is more significant when they are closer to the two corners. The S-line graph represents upregulated metabolites, the substance above the horizontal axis represents upregulated metabolites, and the substance below represents downregulated metabolites (Figure 3). The darker the red color, the greater the difference in the metabolite [20,21,22,23]. Consistently, the OPLS-DA model was effective and had good predictive ability, which means that it could be used for further analysis.

In order to better compare the level of the differences of the metabolites between any two groups, a volcanic map was plotted (FC ≥ 2, *p* < 0.05) (Figure 4). The horizontal axis represents multiple differences and the vertical axis represents the significance. The substance on the left side of the volcanic map axis represents the downregulated differential, while the right side represents the upregulated differential. In positive ion mode, the MC vs. NC group, the CFD-L vs. MC group, the CFD-M vs. MC group, and the CFD-H vs. MC group, filtered out 91, 152, 109, and 173 differential metabolites, respectively. In negative ion mode, the MC vs. NC group, the CFD-L vs. MC group, the CFD-M vs. MC group, and the CFD-H vs. MC group, filtered out 285, 200, 260, and 243 differential metabolites, respectively (Figure 4).

The VIP values were obtained through the OPLS-DA model. It measured the impact intensity of various metabolite expression patterns [23]. In the VIP plot, metabolites with values higher than 1 were significantly different metabolites [24]. A total of 93 differential metabolites were identified through VIP screening of the OPLS-DA model (VIP > 1, *p* < 0.05) and the volcanic plots (Table 1). These metabolites exhibited certain discriminatory abilities in distinguishing the MC, NC, and CFD groups. However, their accuracy in predicting constipation occurrence and monitoring treatment efficacy still needs to be validated through more research.

### 2.3. The Effect of the CFD on Metabolic Pathways in Constipated Mice 

The MetaboAnalyst 5.0 data analysis software was employed to perform enrichment pathway analysis for the previously screened differential metabolites. The horizontal axis of the KEGG pathway enrichment bubble chart represents the degree of enrichment of the differential metabolites within the pathway and a higher value indicates a greater degree of enrichment. The color intensity of each point correlates with its *p*-value, with darker colors indicating smaller *p*-values and, thus, greater statistical significance. The size of each point reflects the quantity of the differential metabolites present in the pathway. The pathways according to which the CFD alleviated the impact of constipation in regard to the metabolites mainly involved glycerophospholipid biosynthesis; aspartate, alanine, and glutamate metabolism; tryptophan metabolism; the citrate cycle (TCA cycle); arginine biosynthesis; tyrosine metabolism; pyruvate metabolism; leucine, valine, and isoleucine degradation; as well as arginine and proline metabolism (Figure 5). Based on previous research, the CFD significantly improved the serum levels of gastrointestinal regulatory peptides, increased short-chain fatty acid (SCFA) content, and reduced colon damage. Additionally, the CFD significantly increased the mRNA levels of AQP3, AQP9, SCF, and c-Kit, as well as the expression levels of their corresponding proteins. The fecal microbiota results showed that the CFD group significantly increased species richness. In addition, the CFD increased the abundance of potential beneficial bacteria and reduced the number of potential pathogenic bacteria [15]. Therefore, by combining serum metabolism and searching for differential metabolites and pathways in the KEGG database, the possible metabolic mechanism of the CFD in regard to the alleviation of constipation induced by loperamide in mice was proposed (Figure 6).

Through an intergroup comparison, 9,10-Epoxy-18-hydroxystearate and Octadecanoic acid were identified as differential metabolites between the MC and NC groups, which may be attributed to the metabolic effects induced by loperamide. In the comparison between the MC and CFD groups, fumarate, phosphatidylethanolamine, L-Norleucine, and N-Acetyl-L-aspartate, among other differential metabolites, demonstrated the effects of the CFD on alleviating constipation. Therefore, to better analyze the metabolic effects induced by the CFD, the subsequent sections include a discussion of the relevant metabolic pathways in detail.

#### 2.3.1. TCA Cycle

Compared with the mice in the normal group, the energy metabolism level of intestinal smooth muscle cells in constipated mice was reduced, which led to a gastrointestinal motility deficiency, abnormal TCA circulation, and energy metabolism disorders [25]. Fumarate is an organic compound widely present in the bodies of animals and plants. It may promote intestinal peristalsis by affecting the intestinal nervous system or altering the pH value of the intestinal environment, thereby helping to alleviate constipation. Moreover, it may alter the microbial environment within the intestine, affecting the balance of the gut microbiota. As a key intermediate in the TCA cycle, it has analgesic effects, improves blood circulation, and inhibits platelet activation. Its anti-inflammatory effect was manifested in inhibiting leukocyte chemotaxis and phagocytosis, reducing capillary permeability, and inhibiting histamine release. After treatment with the CFD, the relative content of fumarate increased, indicating that metabolism recovered and the CFD effectively improved constipation in mice.

#### 2.3.2. Glycerophospholipid Biosynthesis

This study found that the pathways responsible for the metabolism of sphingolipids and glycerophospholipids were the most notable pathways involved in the relevant process, similar to Zhang’s findings [26]. Phosphatidylethanolamine (PE) is a rich type of glycerol phospholipid. In living organisms, PE is hydrolyzed by phospholipase into free fatty acids, which undergoes β-oxidation and provides ATP as an important energy source [27]. It was reported that glycerophospholipid metabolism plays a crucial role in the inflammatory network [28]. Research has found that abnormal PE abundance and disrupted glycerophospholipid metabolism were essential factors in the development of malignant tumors in patients with cancer, such as colorectal cancer [29]. It showed that the relative abundance of PE decreased after CFD treatment, indicating that CFD treatment alleviated the disorder of glycerophospholipid metabolism and reduced the occurrence of inflammation. 

#### 2.3.3. Amino Acid Metabolism

This study found that the alleviation of constipation by the CFD was closely related to its regulation of energy metabolism and amino acid metabolism. The content of N-Acetyl-L-aspartate, L-Norleucine, and cis-4-Hydroxy-D-proline significantly increased, thereby promoting the synthesis of proteins related to rapid epithelial turnover and mucin production. Amino acid metabolism provided energy and promoted intestinal peristalsis, and synthesized proteins, amino acids, and purines, etc. Meanwhile, amino acid metabolism provided nitrogen and carbon sources for intestinal mucosal cells and intestinal microbiota, and maintained the number of intestinal mucosal cells and regulated the gut microbiota environment [30], especially N-Acetyl-L-aspartate and Hydroxyproline. Aspartate helped to produce several other amino acids (including asparagine, arginine, and lysine) and Hydroxyproline was a protective substance and free radical scavenger for the body’s inner membrane and enzymes. They enhanced the functionality of the intestinal barrier and upregulated the expression of anti-inflammatory cytokines and tight junction proteins. This intervention diminished oxidative stress and apoptosis of intestinal cells during inflammation and inhibited the expression of pro-inflammatory factors, thereby contributing to the alleviation of constipation [31]. In summary, the CFD promoted intestinal motility, improved the intestinal barrier, enhanced immunity, and alleviated constipation by regulating metabolic disorders, such as energy metabolism and amino acid metabolism. However, the pathway mechanism of CFD treatment efficacy is currently only based on a speculative analysis, and further basic and clinical research is needed to verify these hypotheses.

## 3. Materials and Methods

### 3.1. Preparation of the CFD

A CFD was obtained based on the previous method detailed in [32]. Briefly, fruit powders (including 24% blueberry, 11% dragon fruit, 20% yacon, 13% lemon, 13% purple sweet potato, and 19% papaya) were dissolved in water. Chinese herbs (with a mass fraction of 16% licorice, 50% tangerine peel, 13% honeysuckle, and 21% Poria cocos) wrapped in gauze were placed in boiling water for 30 min. Subsequently, the traditional Chinese medicine solution was mixed with the fruit powder solution and oligofructose, sucralose, and steviol glycoside were added until a constant volume of 1 L was reached. Then, the mixture was centrifuged (8000× *g*, 15 min) and filtered with a 1.2 µm microporous filter membrane. Finally, the CFD was obtained following high-pressure steam sterilization at 115 °C for 30 min.

### 3.2. Animal Experimental Design and Sample Collection

The licensing information for the female pathogen-free KM mice (20 ± 2 g) (Slake Jingda Animal Co., Ltd., Changsha, Hunan, China) is SCXK (Xiang) 2019-0004. These mice were acclimated to a controlled environment (23 ± 1 °C, with a 12-h light/dark cycle) for one week. The animal experiments were approved by the Animal Ethics Committee at Hainan University (No. HNUAUCC2021–00118).

Following adaptive feeding, the mice were randomly assigned to the following groups: normal control (NC); model control (MC); and low, medium, and high-dose CFD groups (CFD-L, CFD-M, CFD-H). Within a week, the NC group mice were given physiological saline by gavage administration daily, while the other mice were given loperamide by gavage administration (10 mg/kg body weight (bw)) to establish a constipation model. After modeling, the MC group was orally administered with physiological saline every day. The low, medium, and high-dose CFD groups were gavaged with fluids containing 0.015, 0.03, and 0.06 mL/g bw of the CFD, respectively.

On the final day of the experiment, the mice fasted for 12 h, their orbital blood was collected, and it was left to stand for 1 h. The collected blood was centrifuged (4000× *g*, 15 min) to obtain serum. The serum sample was mixed with methanol extract in a ratio of 1:3. Then, the solution was centrifuged (12,000× *g*, 10 min, 4 °C). The supernatant was collected and filtered, using a 0.45 μm organic membrane.

### 3.3. Chromatographic and Mass Spectrometry Conditions

This experiment follows the method by Yang et al., with slight modifications [16]. High-performance liquid chromatography was carried out using the ExionLC system (ExionLC AD, SCIEX, Framingham, MA, USA), equipped with an Agilent Zorbax Eclipse Plus C18 chromatography column (3.0 mm × 150 mm, 1.8 μm). The mobile phase involved a 0.1% formic acid aqueous solution (A) and acetonitrile (B). The elution program was for 0–1.5 min, 5% B; 1.5–15 min, 5–60% B; 15–25 min, 60–100% B; 10–18 min, 52–65% B; 30–30.10 min, 100–5% B; 30.1–35 min, 5% B. The column temperature was 35 °C, and the injection volume was 3 μL. The mass spectrometry was run in positive and negative ion modes, with a range of 100–1200 Da, and the temperature was 325 °C. The voltage was 140 V and the eighth level RF voltage was 750 V. The sheath gas was nitrogen. 

### 3.4. Analysis of Biomarkers

The data was subjected to multivariate statistical analysis using SIMCA-P software 14.0. An orthogonal partial least squares discriminant analysis (OPLS-DA) model was used to screen for potential biomarkers (VIP > 1, *p* < 0.05) and the preliminary screening results are included in the Table 1. Small-molecule substances with distinctive characteristics were identified through searches and manual verification. MetaboAnalyst 5.0 and the KEGG database were used for pathway analysis of potential labeled metabolites, to elucidate the molecular mechanisms of the metabolic changes.

### 3.5. Metabolomics and Statistical Analysis

OriginPro 2018, GraphPad Prism 8, SIMCA-P (Umetrics, Umeå, Sweden), and the MetaboAnalyst 5.0 database were used for plotting and data analysis. *p* < 0.05 was considered statistically significant.

## 4. Conclusions

The effects of a CFD on serum metabolism in constipation mice were studied using UHPLC–QTOF/MS. The PCA and OPLS-DA models showed that each group was relatively isolated, indicating that the CFD had a significant effect on the metabolic substances in constipated mice, and 93 differential metabolites were identified. Through the use of the KEGG database and MetaboAnalyst 5.0 analysis, it was found that the pathways in which compound beverages alleviate constipation and affect metabolites mainly involve nine metabolic pathways. There was a significant difference in the metabolite levels between loperamide-induced constipated mice and normal mice, resulting in the disruption of metabolic pathways, such as the TCA cycle, pyruvate metabolism, and tyrosine metabolism, in the body. After CFD treatment, the intestinal environment was improved and the metabolic pathway disorders were alleviated in mice.

This study investigated the role of a CFD, obtained through the combination of fruits and Chinese herbs, in regulating constipation symptoms in mice, as well as explored the metabolic pathways in terms of mouse serum. It has provided preliminary insights into the potential mechanisms of CFDs in the treatment of constipation and has laid the foundation for future research. The findings are of great significance in promoting the comprehensive utilization of food and medicine homologous components and the development of compound functional beverages.

## Figures and Tables

**Figure 1 ijms-26-00702-f001:**
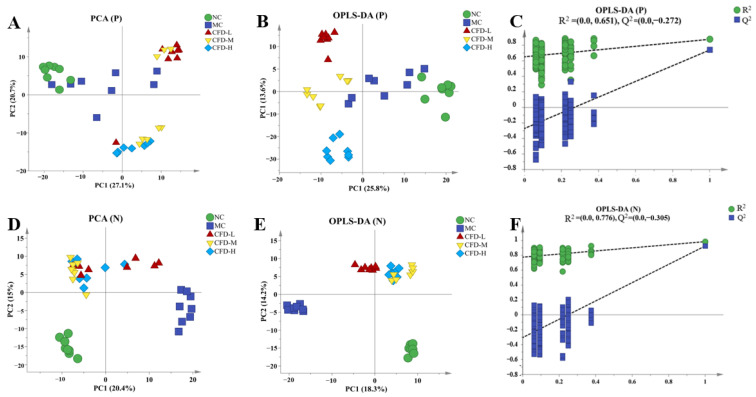
The PCA score plots (**A**,**D**), OPLS-DA score plots (**B**,**E**), and the validation model (**C**,**F**) in positive and negative ion modes for the serum metabolites.

**Figure 2 ijms-26-00702-f002:**
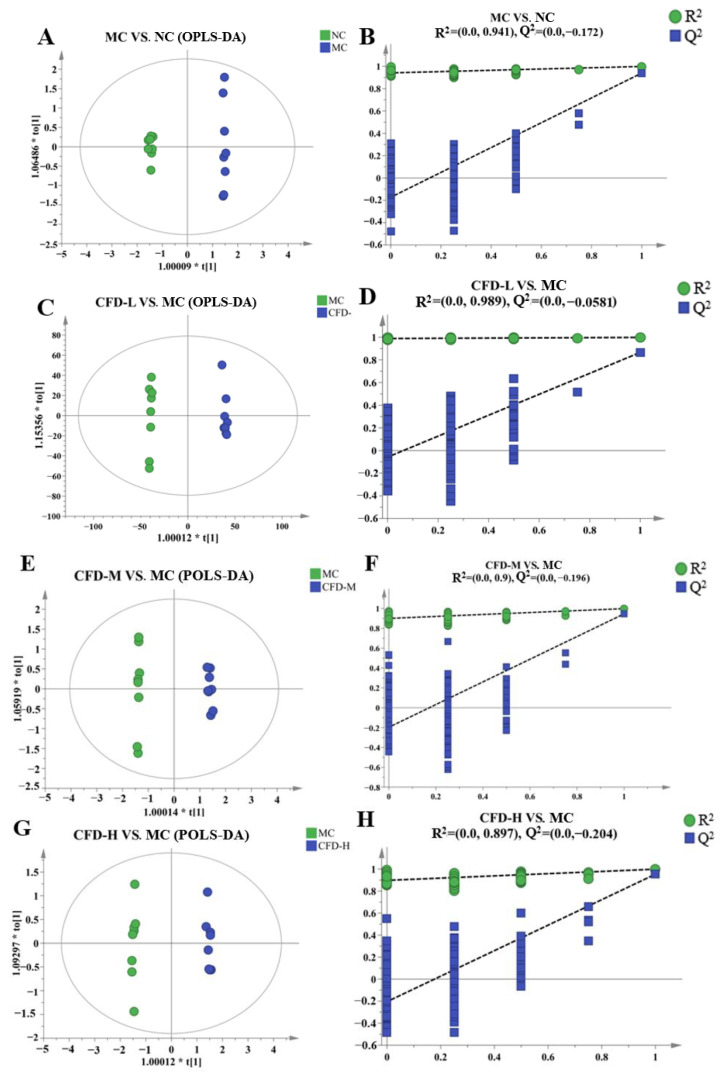
OPLS-DA score plot (**A**,**C**,**E**,**G**) and validation model (**B**,**D**,**F**,**H**) for comparison of metabolite differences in different groups.

**Figure 3 ijms-26-00702-f003:**
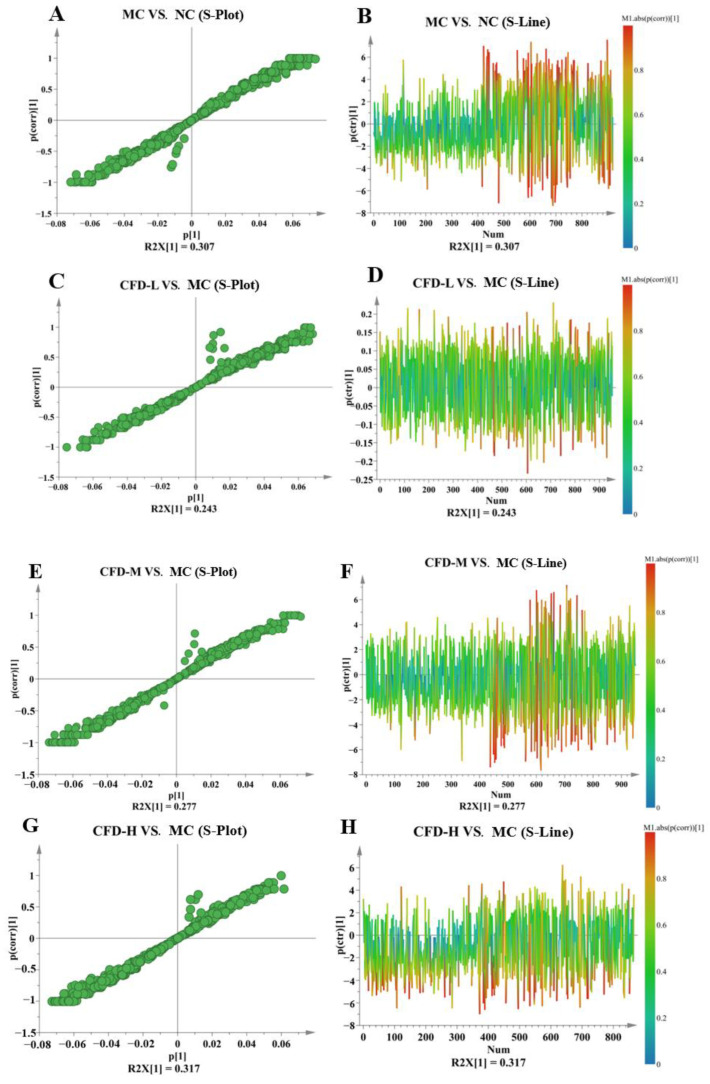
S-plot (**A**,**C**,**E**,**G**) and S-line (**B**,**D**,**F**,**H**) for comparison of metabolite differences in different groups.

**Figure 4 ijms-26-00702-f004:**
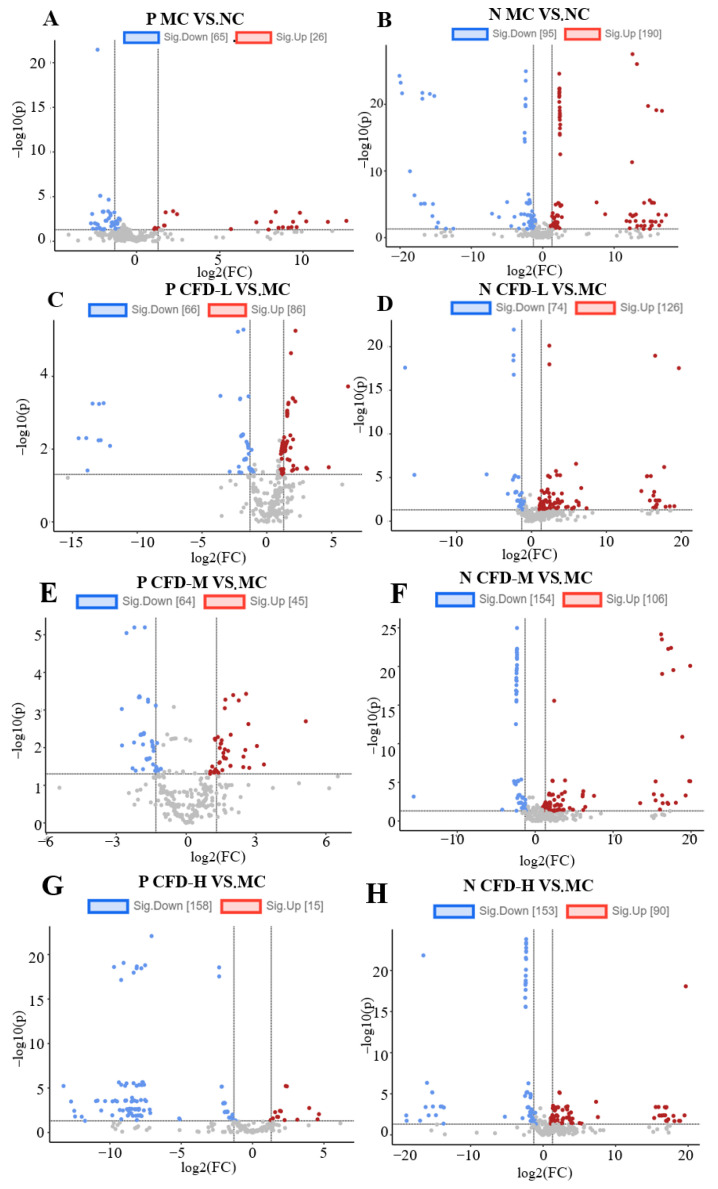
Volcanic plot analysis in positive ion mode (**A**,**C**,**E**,**G**) and negative ion mode (**B**,**D**,**F**,**H**) of the impact of the compound drink on metabolites.

**Figure 5 ijms-26-00702-f005:**
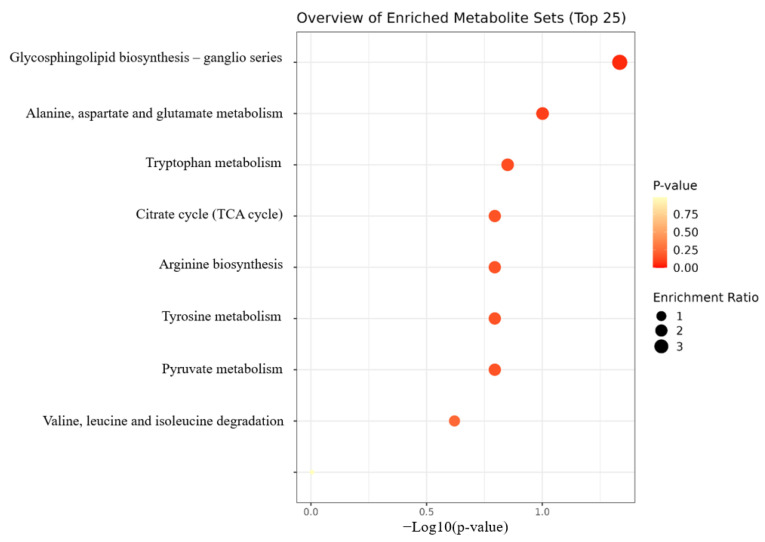
Differential metabolite pathway analysis.

**Figure 6 ijms-26-00702-f006:**
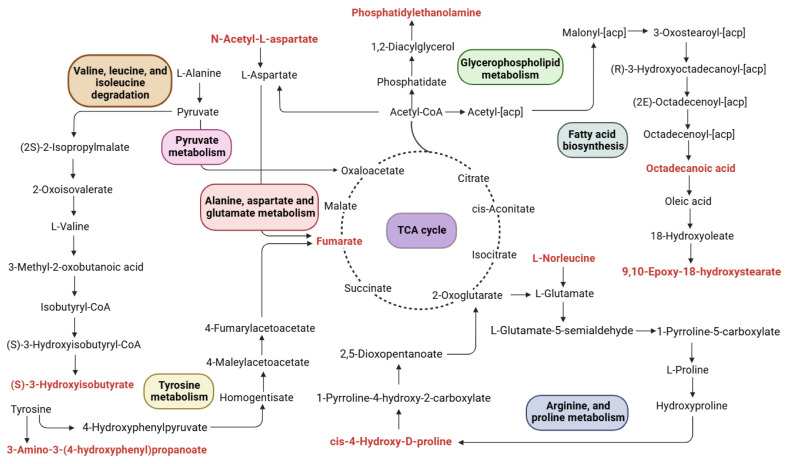
Metabolite effects of compound drink on mice with constipation.

**Table 1 ijms-26-00702-t001:** Differential metabolites analysis of serum between groups.

NO.	Compound	Ionization Model	Formula	HMDB ID	MC/NC		CFD-H/MC	
					Log FC	Change	Log FC	Change
1	3,3-Difluoro-17-methyl-5alpha-androstan 17beta-ol	+	C20H32F2O		4.4505	up	−3.218392	down
2	4’-O-Methylneobavaisoflavone 7-O-(2’’-p coumaroylglucoside)	+	C36H36O11		2.7664113	up	−2.766411	down
3	6-Hydroxydelphinidin 3-(6 malonylglucoside)	+	C24H23O16		2.4042294	up	−14.65394	down
4	Ala Arg	+	C9H19N5O3		3.0752976	up	−3.075298	down
5	Ammothamnidin	+	C25H28O5		3.5802338	up	−1.706038	down
6	Anabasine	+	C10H14N2	HMDB04350	−8.89981	down	2.7541528	up
7	Arg Lys Asp	+	C16H31N7O6		−3.9354963	down	1.4209375	up
8	Asp Tyr Gln	+	C18H24N4O8		3.1950102	up	−1.556805	down
9	Benzo[ghi]fluoranthene	+	C18H10		3.30509	up	−3.30509	down
10	Boschniakine	+	C10H11NO		−6.325095	down	3.3662586	up
11	C.I Orange G	+	C16H12N2O7S2		−7.0338855	down	1.3723431	up
12	Carbamorph	+	C8H16N2OS2		3.253957	up	−1.591061	down
13	Cyclic adenosine diphosphate ribose	+	C15H21N5O13P2		−4.64132	down	4.572323	up
14	DG(17:0/22:1(13Z)/0:0)	+	C42H80O5		−3.099382	down	6.0656075	up
15	Dimethylenetriurea	+	C5H12N6O3		4.5084496	up	−2.203515	down
16	FTY720 phenoxy-biotin	+	C27H44N4O5S		1.5744886	up	−1.574489	down
17	Lys Lys Lys	+	C18H38N6O4		−5.022828	down	4.394418	up
18	Mephobarbital	+	C13H14N2O3		1.2537704	up	−1.25377	down
19	Myricanene A 5-[arabinosyl-(1-6) glucoside]	+	C32H42O13	HMDB39351	1.4820877	up	−1.482088	down
20	Oxolucidine B	+	C30H49N3O2		2.976674	up	−13.87485	down
21	Pachymic acid	+	C33H52O5		−3.5755281	down	1.7724607	up
22	PS(20:4(5Z,8Z,11Z,14Z)/21:0)	+	C47H84NO10P		−3.8085504	down	6.5558763	up
23	Quercetagetin 4’-methyl ether 7-(6-(E) caffeylglucoside)	+	C31H28O16		2.6961927	up	−1.310424	down
24	Ssioriside	+	C27H38O12	HMDB38934	1.4357603	up	−1.404921	down
25	Tetradecanoylcarnitine	+	C21H42NO4	HMDB05066	2.4656892	up	−3.78558	down
26	Tyr Val	+	C14H20N2O4		−1.6964296	down	4.730801	up
27	Val Ile Leu	+	C17H33N3O4		5.294032	up	−10.76906	down
28	(S)-3-Hydroxyisobutyrate	−	C4H8O3		−1.8594704	down	4.038164	up
29	(+)-trans-alpha-Irone	−	C14H22O		−1.5312119	down	1.5506911	up
30	(9S,13S)-1a,1b-dihomo-jasmonic acid	−	C14H22O3		−5.3799334	down	8.763475	up
31	(3a,5b)-24-oxo-24-[(2 sulfoethyl)amino]cholan-3-yl-b-D Glucopyranosiduronic acid	−	C32H53NO11S	HMDB02429	11.074536	up	−11.07454	down
32	L-Norleucine	−	C6H13NO2	HMDB01645	−1.8873906	down	4.8549323	up
33	(R)-Pantolactone	−	C6H10O3		−4.170607	down	9.963216	up
34	1,8-Naphthyridine-3-carboxylic acid, 1 ethyl-1,4-dihydro-7-hydroxy-4-oxo-	−	C11H10N2O4		−2.956582	down	6.1091447	up
35	cis-4-Hydroxy-D-proline	−	C5H9NO3		−2.724896	down	5.8799667	up
36	10-Deoxygeniposide tetraacetate	−	C25H32O13		10.379653	up	−10.37965	down
37	11-Hydroxyprogesterone 11-glucuronide	−	C27H38O9		16.0126	up	−16.0126	down
38	1-O-[(6’-O-hexadecanoyl)-a-D glucopyranosyl]-(2-hexadecanoyloxy) eicosan-1-ol	−	C58H112O9		12.955203	up	−8.445993	down
39	1-Octen-3-yl glucoside	−	C14H26O6	HMDB32959	−3.689945	down	4.358061	up
40	2-[[(3a,5b,7b)-7-hydroxy-24-oxo-3 (sulfooxy)cholan-24-yl]amino] Ethanesulfonic acid	−	C26H45NO9S2	HMDB02449	16.910694	up	−14.94277	down
41	2-Chloro-1,1,2-trifluoroethyl ethyl ether	−	C4H6ClF3O		−4.176031	down	3.1676066	up
42	2-oxo-tetradecanoic acid	−	C14H26O3		1.2939825	up	−11.21112	down
43	3-Amino-3-(4-hydroxyphenyl)propanoate	−	C9H11NO3	HMDB03831	3.940454	up	−5.338254	down
44	3-Isochromanone	−	C9H8O2		−2.7620492	down	5.775576	up
45	5-Hydroxydantrolene	−	C14H10N4O6	HMDB60776	−2.713782	down	7.952751	up
46	9,10-Epoxy-18-hydroxystearate	−	C18H34O4		−6.57066	down	5.5488477	up
47	Anhwiedelphinine	−	C35H44N2O10		12.642545	up	−12.64255	down
48	Arg Phe Arg	−	C21H35N9O4		16.213037	up	−14.42369	down
49	Armillatin	−	C38H58O6	HMDB38743	−6.374333	down	8.174591	up
50	Aromatized deshydroxy-C-1027 chromophore	−	C43H44ClN3O12		10.24949	up	−8.825224	down
51	Auriculoside	−	C22H26O10		12.162231	up	−12.16223	down
52	Bambuterol	−	C18H29N3O5	HMDB15478	−1.3087604	down	2.2535157	up
53	beta-D-Mannosylphosphodecaprenol	−	C56H93O9P		−16.54321	down	4.239726	up
54	Broussoflavonol D	−	C30H32O7		13.831339	up	−13.83134	down
55	Bufotalin	−	C26H36O6		−10.536349	down	10.325614	up
56	Chlorpromazine sulfone	−	C17H19ClN2O2S		−5.340466	down	10.195431	up
57	Citranaxanthin	−	C33H44O	HMDB36883	13.942801	up	−13.9428	down
58	Cromakalim	−	C16H18N2O3		−3.9542727	down	10.889442	up
59	Cypridina luciferin	−	C22H27N7O		15.838913	up	−8.986485	down
60	decanamide	−	C10H21NO		1.2328752	up	−8.027856	down
61	Fenoterol sulfate	−	C17H21NO7S		2.8744664	up	−3.97058	down
62	Fexaramine	−	C32H36N2O3		9.810449	up	−9.810449	down
63	Fumaric acid	−	C4H4O4	HMDB00134	−4.888547	down	3.24939	up
64	Ganglioside GA2 (d18:1/12:0)	−	C50H92N2O18	HMDB04888	13.875375	up	−13.87538	down
65	Gingerglycolipid A	−	C33H56O14	HMDB41093	−1.4199212	down	4.305419	up
66	Gingerol	−	C17H26O4	HMDB05783	9.376551	up	−9.376551	down
67	Ginkgolide J	−	C20H24O10		9.069026	up	−6.129434	down
68	Gliadin	−	C29H41N7O9	HMDB34486	12.318045	up	−13.5362	down
69	Glu Trp Ala	−	C19H24N4O6		9.855809	up	−11.23999	down
70	Helilupolone	−	C30H38O4		−9.420137	down	5.3089356	up
71	His-Phe-OH	−	C21H20N4O6		−1.8027265	down	4.804309	up
72	Kobusone	−	C14H22O2		−1.4395556	down	3.3383055	up
73	Lauroyl diethanolamide	−	C16H33NO3	HMDB32358	−5.614107	down	5.2017584	up
74	Leu Leu Phe	−	C21H33N3O4		12.384389	up	−10.56313	down
75	Maraviroc	−	C29H41F2N5O	HMDB15584	−1.3437376	down	4.8805656	up
76	Menthol propylene glycol carbonate	−	C14H26O4	HMDB39785	16.521048	up	−1.216217	down
77	Notoginsenoside I	−	C54H92O22	HMDB31371	7.2977753	up	−1.388841	down
78	Octadecanoic acid	−	C19H36O2		11.138865	up	−9.72685	down
79	Patuletin 3-rhamnoside-7-(4” acetylrhamnoside)	−	C30H34O17		12.763924	up	−2.99884	down
80	PE(18:4(6Z,9Z,12Z,15Z)/20:2(11Z,14Z))	−	C43H74NO8P	HMDB09198	−1.393867	down	10.985636	up
81	Perindopril lactam	−	C19H30N2O4		2.6985683	up	−2.698568	down
82	Phe Lys Trp	−	C26H33N5O4		−11.943593	down	7.563178	up
83	PI(22:0/20:0)	−	C51H99O13P		−12.499487	down	4.0756774	up
84	PI(22:4(7Z,10Z,13Z,16Z)/21:0)	−	C52H93O13P		4.5378876	up	−4.537888	down
85	PI(P-20:0/15:0)	−	C44H85O12P		15.159404	up	−15.1594	down
86	PI-Cer(d18:0/16:0)	−	C40H80NO11P		15.341745	up	−15.34175	down
87	Prieuranin acetate	−	C40H52O17		−9.76871	down	8.086171	up
88	Rofecoxib	−	C17H14O4S		−1.2988684	down	12.447737	up
89	Saphenamycin	−	C23H18N2O5		11.704618	up	−1.794679	down
90	Tamsulosin	−	C20H28N2O5S	HMDB14844	10.714832	up	−15.24225	down
91	TG(17:2(9Z,12Z)/20:1(11Z)/22:1(11Z))	−	C62H112O6		−15.996837	down	4.5640464	up
92	TG(20:4(5Z,8Z,11Z,14Z)/20:5(5Z,8Z,11Z,1 4Z,17Z)/22:2(13Z,16Z))	−	C65H104O6		12.925481	up	−4.356386	down
93	Trabectedin	−	C39H43N3O11S		9.429379	up	−6.670994	down

## Data Availability

Data will be made available on request.

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
