# Peer review of "Mechanism of Functional Compound Fruit Drinks in Regulating Serum Metabolism in Constipated Mice"

_ijms, 2025, doi:10.3390/ijms26020702_

Round 1

Reviewer 1 Report

Comments and Suggestions for Authors

Comments for ijms-3380891

This study investigates the alleviating effects of a compound fruit drink (CFD) on constipation in a loperamide-induced mouse model, focusing on its regulatory influence on serum metabolism. Utilizing advanced ultra-high-performance liquid chromatography-quadrupole time-of-flight mass spectrometry (UHPLC-Q-TOF-MS), the research identified 93 differential metabolites, including phosphoglycerides and amino acid derivatives. Pathway enrichment analysis revealed significant alterations in metabolic pathways such as glycerophospholipid metabolism, the tricarboxylic acid (TCA) cycle, pyruvate metabolism, and tyrosine metabolism. These findings demonstrate CFD’s ability to restore metabolic balance, improve the intestinal environment, and enhance gastrointestinal motility. This study offers promising insights for the development of functional foods targeting gut health and constipation alleviation. However, the study has several limitations that warrant discussion.

1.       The use of a loperamide-induced constipation model may not fully capture the multifactorial etiology of constipation observed in human populations.

2.       The study’s small sample size may limit the statistical power and generalizability of the findings.

3.       The research focused on short-term effects, leaving questions about the long-term safety and efficacy of CFD unanswered.

4.       While the animal model provides valuable preliminary data, human trials are necessary to translate these findings into practical applications.

5.       Although metabolic pathways were identified, the study lacked in-depth exploration of specific molecular mechanisms underlying CFD’s effects.

6.       CFD’s formulation includes multiple fruit and herbal components, making it challenging to attribute observed effects to specific ingredients.

7.       The study did not account for potential interactions between CFD and dietary or lifestyle variables, which are critical in managing constipation.

Author Response

Dear Editors and Reviewers,

Thank you for your kindly and detailed comments concerning our manuscript (IJMS-3380891). Those comments are valuable and very helpful for revising and improving our paper, as well as the important guiding significance to our further research. We have studied comments carefully and revised this manuscript according to the comments and suggestions of editors and reviewers. The corrections were marked in red in the paper. The main corrections in the paper and the details of responses to the comments were shown as follows.

Response to the Editor’s Comments:

We would like to thank the reviewer for her/his careful revision. All the comments have been appreciated and addressed in the manuscript highlighted in red. In the following we answer all comments/questions point by point.

Comments 1: The use of a loperamide-induced constipation model may not fully capture the multifactorial etiology of constipation observed in human populations.

Response 1: Thank you for your careful review of our research. We fully agree with your viewpoint. Indeed, the constipation model induced by loperamide mainly simulates drug-induced constipation, which may differ from the observed multifactorial causes of constipation in the population. We acknowledge that this model does not cover all types of constipation, especially those caused by multiple factors (such as age, diet, lifestyle, mental status, intestinal flora imbalance, etc.). However, for existing articles on constipation, the loperamide model is still the main way to explore the effects of constipation [1-3]. Although the model has its limitations, we believe that this study provides important insights into the mechanism of constipation caused by loperamide and provides a scientific basis for the development of compound functional beverages that can regulate constipation.

In order to comprehensively capture the multifactorial causes of constipation, we may adopt the following strategies in the future: combining multiple constipation models, including surgical induction, dietary induction, etc., to simulate different constipation situations. Using systems biology methods, investigate the interactions between gut microbiota, genetic factors, and environmental factors.

  1. Lee, H.-Y.; Kamal Hossain, M.; Kim, S.-H.; Jeong, P.-Y.; Lee, G.-H.; Kim, D.-S.; Ja Chung, M.; Chae, H.-J., Improving intestinal health and mitigating Loperamide-Induced constipation through the modulation of Aquaporin-3 expression, reduction of oxidative stress, and suppression of inflammatory response by fermented rice extract. Journal of Functional Foods 2024, 121, 106444.
  2. Jiang, L.; Zhang, R.; Lin, X.; Tuo, Y.; Mu, G.; Jiang, S., The preparation of synbiotic AHY relieving loperamide-induced constipation and its modulation mechanism in vivo. Food Bioscience 2024, 59, 104096.
  3. Cai, W.-F.; Lin, S.-X.; Ma, P.-Y.; Shen, C.-Y., Semen Pruni oil attenuates loperamide-induced constipation in mice by regulating neurotransmitters, oxidative stress and inflammatory response. Journal of Functional Foods 2023, 107, 105676.

Comments 2: The study’s small sample size may limit the statistical power and generalizability of the findings.

Response 2: Thank you for your careful review of our research and for pointing out that sample size may limit the statistical power and generalizability of our findings. We agree with your concerns and would like to explain our decision and discuss its impact on the research results. Our research is based on the results of pilot experiments, similar studies in existing literature, and funding limitations regarding the selection of sample size. After one week of adaptive feeding, mice were randomly divided into six groups (n=10), and 8 mice were randomly selected for the experiment, which met the experimental requirements. This study provides preliminary evidence that the compounded beverage relieves constipation and provides hypotheses and direction for further large-scale studies.

Comments 3: The research focused on short-term effects, leaving questions about the long-term safety and efficacy of CFD unanswered.

Response 3: Thank you very much for your evaluation of our research. You correctly pointed out that our research mainly focuses on the short-term effects of CFD. We chose this time frame based on the following considerations:

  1. Preliminary evidence: Our goal is to provide preliminary evidence to confirm the short-term effects of CFD on specific biomarkers or physiological processes, which is the basis for further research.
  2. Resource limitations: Due to funding and time constraints, we were unable to conduct long-term tracking research at this stage.
  3. Research Design: Short term research design helps us control experimental variables, reduce potential data variability, and thus observe the direct effects of CFD more clearly.

We recognize that long-term safety and efficacy are important aspects in evaluating any treatment method. Long term research can reveal the long-term effects, potential side effects, and safety issues of treatment measures after long-term use. To address this limitation, we emphasized the necessity of long-term research in the discussion section and proposed the following suggestions: We plan to conduct long-term tracking in future studies to evaluate the long-term effects and potential adverse reactions of CFD. Conduct long-term CFD treatment studies in animal models to evaluate their long-term safety and potential chronic effects. We believe that through these future studies, we can gain a more comprehensive understanding of the safety and efficacy of CFD, and provide stronger evidence for clinical practice. We appreciate your valuable feedback.

Comments 4: While the animal model provides valuable preliminary data, human trials are necessary to translate these findings into practical applications.

Response 4: Thank you for your evaluation of our research and suggestions for further human trials. We fully agree with your point of view that although animal models provide us with valuable preliminary data, human trials are indeed needed to translate these findings into practical applications. Animal models play a crucial role in revealing potential biological mechanisms, evaluating the safety of treatments, and providing preliminary efficacy data. Through animal experiments, we can understand the role of compound functional beverages in regulating constipation, and lay a solid foundation for the future application of compound functional beverages in human beings to alleviate constipation.

Comments 5: Although metabolic pathways were identified, the study lacked in-depth exploration of specific molecular mechanisms underlying CFD’s effects.

Response 5: Thank you for your careful review of our research and your valuable feedback. Our research mainly focuses on determining the impact of CFD on metabolic pathways, as we believe this is the foundation for understanding its potential health benefits. In Figure 6, we focused on explaining the molecular mechanism of CFD action. This study further speculates on the metabolic pathways of CFD by identifying biomarkers such as Fumarate, Phosphatidylethanolamine and N-acetyl-Laspartate. We analyzed these biomarkers, such as the increase in fumaric acid content enhances its anti-inflammatory effect, manifested by inhibiting leukocyte chemotaxis and phagocytosis, reducing capillary permeability, inhibiting histamine release, and regulating constipation. By identifying these pathways, we hope to provide direction and theoretical basis for future research.

Comments 6: CFD’s formulation includes multiple fruit and herbal components, making it challenging to attribute observed effects to specific ingredients.

Response 6: Thank you for your constructive feedback. We can’t determine which specific component is responsible for regulating constipation. However, this article developed a composite functional drink with the aim of improving the comprehensive utilization of fruits and traditional herbs, as well as reducing excessive intake of a single ingredient by adjusting the ratio of fruits to Chinese herbs, and developing a beverage with the effect of regulating constipation. Further explore the regulatory and laxative effects of this composite functional drink on constipated mice. To clarify the specific functions of each component, more targeted experiments are needed. It may be the direction for future research.

Comments 7: The study did not account for potential interactions between CFD and dietary or lifestyle variables, which are critical in managing constipation.

Response 7: Thanks for your comment. In this experiment, all mice were fed and tested in the same living environment. We strictly controlled the influencing factors of all experimental samples to maintain consistency, and conducted feeding, dissection, and other experiments at the same time. We will adopt a more comprehensive design in future research, including detailed recording and control of individual diet and lifestyle. It helps to better understand the interactions between functional beverages and other behaviors, and contribute to the application of composite functional beverages.

Reviewer 2 Report

Comments and Suggestions for Authors

The article is interesting, and the authors have worked hard to get the results and to prepare the study, but some changes are needed:

- The title should be clarified as it seems more correct like this: "Mechanism of functional compound fruit drink regulating serum metabolism in constipated mice."

- References should be cited in text according to the recommendation -correct line 55.

- I don't recommend the expression "We have explored the alleviating effect." Please modify.

- Probably paragraphs 73-81 represent the aim of the study. Please rephrase to emphasize better the aim of the study.

- Please clarify lines 106-107. "groups were gavaged with BW " ?

- Clarify a little bit: "OPLS-DA model," is an abbreviation for what? line 126 (Orthogonal partial least squares discriminant analysis?)

- Which biomarkers were analyzed? Please detail a little 2.4.

- "back-regulated"? Try to clarify how this is work.

- The metabolites in figure 6 are very nicely presented here. Where are they in your study? A different approach to results presentation is probably necessary.

- Clarify KEGG database—what does it contain?

- The conclusion 

Percent match: 42% is too high? or is it OK for publication?

Author Response

Dear Editors and Reviewers,

Thank you for your kindly and detailed comments concerning our manuscript (IJMS-3380891). Those comments are valuable and very helpful for revising and improving our paper, as well as the important guiding significance to our further research. We have studied comments carefully and revised this manuscript according to the comments and suggestions of editors and reviewers. The corrections were marked in red in the paper. The main corrections in the paper and the details of responses to the comments were shown as follows.

Response to the Editor’s Comments:

We would like to thank the reviewer for her/his careful revision. All the comments have been appreciated and addressed in the manuscript highlighted in red. In the following we answer all comments/questions point by point.

Comments 1: The title should be clarified as it seems more correct like this: "Mechanism of functional compound fruit drink regulating serum metabolism in constipated mice."

Response 1: We do perfectly agree with you and we have revised the title in the manuscript according to your comments (revised in line 3). Thank you for your suggestion, which has helped us improve the clarity and accuracy of our paper.

Comments 2: References should be cited in text according to the recommendation -correct line 55.

Response 2: Thank you very much. We corrected the references in the manuscript (revised in line 66).

Comments 3: I don't recommend the expression "We have explored the alleviating effect." Please modify.

Response 3: Thank you very much for your comment. We rephrased the sentence in the manuscript (revised in line 100-103).

Comments 4: Probably paragraphs 73-81 represent the aim of the study. Please rephrase to emphasize better the aim of the study.

Response 4: We appreciate your comments and have adapted the purpose based on it. We rephrase the aim of the study (revised in line 95-105).

Comments 5: Please clarify lines 106-107. "groups were gavaged with BW "?

Response 5: Thank you for your question. The mice were given loperamide by gavage (10 mg/kg body weight (bw)), and we have revised it in the text (revised in line 132-133).

Comments 6: Clarify a little bit: "OPLS-DA model," is an abbreviation for what? line 126 (Orthogonal partial least squares discriminant analysis?)

Response 6: Thank you for your comment. As you said, OPLS-DA is the Orthogonal partial least squares discriminant analysis, and we clarify it in the text (revised in line 156).

Comments 7: Which biomarkers were analyzed? Please detail a little 2.4.

Response 7: Thank you for your comment. All biomarkers are listed in the supplementary materials table and screened by VIP > 1 and p < 0.05. We mentioned supplementary materials in section 2.4, thank you again for your helpful comments (revised in line 157-158).

Comments 8: "back-regulated"? Try to clarify how this is work.

Response 8: Thanks for your valuable comment. We have carefully reviewed the conclusion of the manuscript and it may not be appropriate to use reverse adjustment here. Therefore, we have deleted this expression in the manuscript (revised in line 200-203).

Comments 9: The metabolites in figure 6 are very nicely presented here. Where are they in your study? A different approach to results presentation is probably necessary.

Response 9: Thank you very much for your feedback. In Figure 6, we focused on explaining the molecular mechanism of CFD action. This study further speculated on the metabolic pathway of CFD by identifying biomarkers such as fumarate, phosphatidylethanolamine, and N-acetyl-L-Aspartic acid. We analyzed these biomarkers, such as an increase in fumaric acid content enhancing its anti-inflammatory effect, and a decrease in phosphatidylethanolamine reducing inflammation. By identifying these pathways, we hope to provide direction and theoretical basis for future research, and in the future, we will adopt new ways to better present this result.

Comments 10: Clarify KEGG database—what does it contain?

Response 10: The KEGG database, also known as the Kyoto Encyclopedia of Genes and Genomes, is a comprehensive biological database that provides key information and powerful analytical tools for numerous research directions in the field of life sciences. Its coverage is extensive and in-depth, and our article mainly includes the KEGG pathway. This is a highly iconic part of the KEGG database, which presents various biochemical metabolic pathways within metabolites in a visual and graphical manner, covering the entire metabolic process of basic substances such as carbohydrates, lipids, and proteins. In these metabolic pathway diagrams, metabolites, enzymes, and the chemical reactions involved in their interconversion are annotated in detail, enabling precise localization of key reaction nodes and key enzymes involved in rate limiting steps. This is crucial for researchers to quickly identify key links in the occurrence of metabolic disorders in disease contexts.

Comments 11: Percent match: 42% is too high? or is it OK for publication?

Response 11: Thanks for your helpful comments. We have reduced overall and individual similarities of the manuscript. Now, the overall similarity has decreased to below 15%.

Round 2

Reviewer 1 Report

Comments and Suggestions for Authors

This article has been greatly improved and the response is substantial.

Author Response

Comment: This article has been greatly improved and the response is substantial.

Response: We would like to express our sincere gratitude to you for the opportunity to revise our manuscript.We greatly appreciate your constructive feedback on our manuscript and the affirmation of our revisions.

Reviewer 2 Report

Comments and Suggestions for Authors

The article now entitled "Mechanism of functional compound fruit drink regulating serum metabolism in constipated mice" has been revised by the authors, and the clarifications made are pertinent and helpful for me to better understand the message of the research. The purpose has been clarified and accurately described.

Abbreviations have been clarified. 

Author Response

Comment: The article now entitled "Mechanism of functional compound fruit drink regulating serum metabolism in constipated mice" has been revised by the authors, and the clarifications made are pertinent and helpful for me to better understand the message of the research. The purpose has been clarified and accurately described.

Abbreviations have been clarified.

Response: Thank you for your positive feedback and recognition. We truly appreciate your thoughtful review and constructive comments. Your support and encouragement are invaluable to us.